# Numerical Covariance Evaluation for Linear Structures Subject to Non-Stationary Random Inputs

M. Domaneschi [1], R. Cucuzza [1,*], L. Sardone [1], S. Londoño Lopez [1], M. Movahedi [2] and G. C. Marano [1]

1. Department of Structural, Geotechnical and Building Engineering, Politecnico di Torino, Corso Duca degli Abruzzi, 24, 10129 Turin, Italy; marco.domaneschi@polito.it (M.D.); laura.sardone@poliba.it (L.S.); santiago.londono@polito.it (S.L.L.); giuseppe.marano@polito.it (G.C.M.)
2. Department of Structural and Geotechnical Engineering, Széchenyi István University, Győr, Hungary
* Correspondence: raffaele.cucuzza@polito.it

**Abstract:** Random vibration analysis is a mathematical tool that offers great advantages in predicting the mechanical response of structural systems subjected to external dynamic loads whose nature is intrinsically stochastic, as in cases of sea waves, wind pressure, and vibrations due to road asperity. Using random vibration analysis is possible, when the input is properly modeled as a stochastic process, to derive pieces of information about the structural response with a high quality (if compared with other tools), especially in terms of reliability prevision. Moreover, the random vibration approach is quite complex in cases of non-linearity cases, as well as for non-stationary inputs, as in cases of seismic events. For non-stationary inputs, the assessment of second-order spectral moments requires resolving the Lyapunov matrix differential equation. In this research, a numerical procedure is proposed, providing an expression of response in the state-space that, to our best knowledge, has not yet been presented in the literature, by using a formal justification in accordance with earthquake input modeled as a modulated white noise with evolutive parameters. The computational efforts are reduced by considering the symmetry feature of the covariance matrix. The adopted approach is applied to analyze a multi-story building, aiming to determine the reliability related to the maximum inter-story displacement surpassing a specified acceptable threshold. The building is presumed to experience seismic input characterized by a non-stationary process in both amplitude and frequency, utilizing a general Kanai–Tajimi earthquake input stationary model. The adopted case study is modeled in the form of a multi-degree-of-freedom plane shear frame system.

**Keywords:** non-stationary random process; covariance analysis; Lyapunov equation; dynamic response and reliability

## 1. Introduction

Taking into account how structures behave in a random vibration setting is a prevalent method used to assess actual response scenarios [1]. This applies to various contexts, e.g., aircraft, vibrating machinery, and buildings subjected to marine or wind vibrations. These engineering scenarios involve examining how structures respond to dynamic and nondeterministic actions, and random dynamic analysis proves to be the most effective mathematical tool for this purpose [2]. This decision arises from the inherent randomness in inputs, a well-documented aspect in the field of random vibration theory (as evidenced by [3–6]). Approaching the problem using these methodologies enables acquiring relevant and reliable information about the structural response, typically unattainable through deterministic methods. The strength of the random vibration approach lies in its high-quality information, including the quantification of structural integrity, which is significant in probabilistic safety assessments. Failure is generally described as the initial moment when a structural crisis begins, tying it to the first instance where one or more measurements of structural response exceed a safe range. This usually involves assessing structural response indicators like displacements, stresses, buckling loads, or natural frequencies.

The random vibration problem for linear mechanical systems subject to Gaussian processes input is posed in stationary environmental conditions as the solution of the so-called Lyapunov matrix equation [7,8] to obtain the response covariance that defines completely the statistics of the system. Challenges in solving the Lyapunov equation have constrained the size of the meshes that could be employed. The Lyapunov equation (i.e., Lyapunov matrix equation, $AR + RA^T + B = 0$, $A$ system matrix, $R$ covariance matrix, and $B$ input matrix, see Section 2) is typically achieved using algorithms like Bartels–Steward or Hessenberg–Schur. These methods require the Schur factorization of system matrix $A$. Various software tools for scientific computing, such as Matlab and Python, employ adapted versions of these algorithms, delivering satisfactory results for small dense matrices $A$ and $B$. This involves O $(N^3)$ floating-point operations and O $(N^2)$ memory [9]. Approaches designed for large system dimensions have been developed, for instance, the Krylov subspace methods [9–11] or the matrix sign function decomposition with Newton's iterative method.

In non-stationary cases, under the same assumptions (linear system and Gaussian input) the time covariance approach is more complex as the Lyapunov matrix covariance becomes a differential one (i.e., Lyapunov differential matrix equation, $\dot{R} = AR + RA^T + B = 0$, see Section 3) whose numerical solution is sometimes more complex, and there are not standard tools to be implemented, differently from the stationary case.

The main objective of this research is to introduce a numerical technique for evaluating response covariance in the time domain for linear structures subjected to non-stationary stochastic loads. This method is tailored for a generic scenario where the input involves a non-stationary modulated filtered white noise process, capable of simulating various real physical loads like earthquakes [12–15].

To achieve a versatile non-stationary approach applicable across different contexts, the structural response is assessed using a covariance approach, as understanding the evolving covariance matrix in the space state is crucial for evaluating reliability, particularly in terms of initial failure events.

To address this, a time-step integration algorithm is proposed, employing the Euler implicit method to solve the differential Lyapunov matrix equation. The outcome is a sequential algorithm that requires the numerical solution of a stationary Lyapunov matrix equation at each time step, a task achievable through standard numerical tools. This method is implemented to reduce computational expenses and has been specifically applied to a multi-story building represented by a shear frame structure, examining dynamic responses under seismic base motions and assessing the reliability concerning initial threshold crossings.

## 2. Linear Elastic MDoF Subject to Non-Stationary Random Vibration

Many instances of real-life structural issues revolve around configurations that match a linear viscoelastic system of lumped masses. These systems face either steady or fluctuating forces. This study considers both scenarios to evaluate how structural responses vary statistically, employing the covariance approach [13,14].

This method presents notable benefits, especially in dynamic conditions, where inputs are simulated as white noises. These can be filtered to better match real dynamic occurrences. By solving the equations of the dynamic equilibrium system, this technique gauges the structural response of a deterministic second-order linear mechanical system composed of lumped masses when subjected to probabilistic dynamic input.

$$\mathbf{M}\overline{\ddot{X}_s}(t) + \mathbf{C}\overline{\dot{X}_s}(t) + \mathbf{K}\overline{X_s}(t) = \mathbf{G}_s\overline{f}(t) \tag{1}$$

where $\overline{X_s}$, $\overline{\dot{X}_s}$ and $\overline{\ddot{X}_s}$ are the structural displacement, velocity, and acceleration process vectors. $\mathbf{M}$, $\mathbf{C}$ and $\mathbf{K}$ are the mass, viscous, and stiffness symmetric matrices. While the

mass matrix is always positive definite, the damping and stiffness matrices are positive semi-definite. The vector

$$\overline{f}(t)^T = [f_1(t), f_2(t), \ldots, f_n(t)] \tag{2}$$

accumulates $n$ stochastic excitations applied to the structure, while $\mathbf{G}s$ represents an $m$ x $n$ matrix linking the excitation components of the forcing vector to the structural degrees of freedom. When the elements of the system excitations vector are stationary white noises, the first- and second-order statistical moments remain unchanged over time.

$$\left\langle f_i^{ST}(t) \right\rangle = \mu_{f_i} \tag{3}$$

$$\left\langle f_i^{ST}(t_1) f_j^{ST}(t_2) \right\rangle = [\mathbf{J}]_{ij} \delta(t_2 - t_1) = \left[ \mathbf{R}_{ff}^{ST}(t_2, t_2) \right]_{ij} \tag{4}$$

Moreover, if $f_i^{ST}(t)$ are Gaussian excitations, then the responses and their time derivatives constitute a Markov vector in the dimension phase state.

The matrix, related to a vector satisfying the shot noise properties (usually denoted as a shot noise vector), has diagonal elements equal to the autocovariance intensity of each force and extra-diagonal elements representing the level of correlation between two generic different forces, so that can vary from if and $f_j$ are completely correlated, to zero, if and are completely un-correlated.

Then, in the case of complete un-correlated forcing loads, the matrix is replaced by the simpler diagonal matrix of components, where the elements are the input power spectral density of each entry.

A commonly used method involves expressing a non-stationary input through an intensity modulation of a stationary process, often referred to as uniform modulation. This method assumes that the intensity of the process alters over time according to a deterministic function $\varphi(t)$, while the spectral contents remain constant. Consequently, in the case of time modulation, a stationary forcing process vector is substituted by the following non-stationary vector:

$$\overline{f}^{NS}(t) = \left[ \varphi_1(t) f_1^{ST}(t), \; \varphi_2(t) f_3^{ST}(t), \ldots, \varphi_n(t) f_n^{ST}(t) \right] \tag{5}$$

with the stochastic characterization

$$\left\langle f_i^{NS}(t) \right\rangle = \varphi_i(t) \mu_{f_i} \tag{6}$$

$$\left\langle f_i^{NS}(t_1) f_j^{NS}(t_2) \right\rangle = \varphi_i(t_1) \varphi_j(t_2) [\mathbf{J}]_{ij} \delta(t_2 - t_1) = \left[ \mathbf{R}_{ff}^{NS}(t_1, t_2) \right]_{i,j} \tag{7}$$

and, in case of un-correlated excitations, the covariance matrix $\mathbf{R}_{ff}^{NS}(t_2, t_2)$ is diagonal

$$\mathbf{R}_{ff}^{NS}(t_1, t_2) = \left\langle f_i^{NS}(t_1) f_j^{NS}(t_2) \right\rangle = \begin{cases} 2\pi S_0^i \varphi_i(t_1) \varphi_i(t_2) \delta(t_2 - t_1) & \text{if } i = j \\ 0 & \text{if } i \neq j \end{cases} \tag{8}$$

*Pre Filters Technique*

In time-domain stochastic analysis, two primary methods are typically employed. The first, applicable when the input's autocorrelation function is known, has been previously outlined. The second involves modeling input processes by solving differential equations using filter techniques, where the input is a white noise process—referred to as the *pre-filter* technique.

The first method is advantageous when the input closely aligns with a shot noise process, providing accurate representation. However, this representation is limited as many real-world phenomena exhibit noticeable frequency modulation, making it suitable only in specific cases.

The second approach is more versatile and capable of representing phenomena with varying frequency contents, even those changing over time. This flexibility is crucial for accurately describing phenomena that could lead to resonant effects in structures. Additionally, this method retains the advantages associated with shot noise inputs, making it the preferred choice for many real structural issues.

In particular, using the *pre-filter* approach, the filter response is described by the $2m_f$ filter space state vector, the solution of the $2m_f$ set of differential equations

$$\overline{\dot{Z}}_f = \mathbf{A}_f(t)\overline{Z}_f + \mathbf{G}_f\overline{W}_f(t) \tag{9}$$

that generally could have a time-dependent form, when not only the frequency but also the amplitude of loads has an intrinsic evolutive nature. $\overline{W}$ is a vector of $n_f$ white noise processes (stationary or non-stationary), $\mathbf{G}_f$ is a $m_{fx}n_f$ matrix that couples the excitation components of the forcing vector to the filter degree of freedom, and finally, $\mathbf{A}_f(t)$ is the $2m_f$ X. $2m_f$ filter system matrix, whose generic form is

$$\mathbf{A}_f(t) = \begin{pmatrix} \mathbf{0} & \mathbf{I} \\ \mathbf{H}_f^1(t) & \mathbf{H}_f^2(t) \end{pmatrix} \tag{10}$$

Then, adopting the *pre-filter* technique, the motion differential equations are written in the space state as

$$\overline{\dot{Z}}_s(t) = \begin{pmatrix} \mathbf{0} & \mathbf{I} \\ -\mathbf{M}^{-1}\mathbf{K} & -\mathbf{M}^{-1}\mathbf{C} \end{pmatrix} \overline{Z}_s(t) + \boldsymbol{\alpha}(t)\overline{Z}_f(t) \tag{11}$$

$$\overline{\dot{Z}}_f = \mathbf{A}_f(t)\overline{Z}_f + \mathbf{G}_f\overline{W}_f(t) \tag{12}$$

where

$$\mathbf{A}_s = \begin{pmatrix} \mathbf{0} & \mathbf{I} \\ \mathbf{H}_s^1 & \mathbf{H}_s^2 \end{pmatrix} \tag{13}$$

is the structural system matrix

$$\boldsymbol{\alpha}(t) = \begin{pmatrix} \mathbf{0} & \mathbf{0} \\ \mathbf{G}_s\boldsymbol{\alpha}_1(t) & \mathbf{G}_s\boldsymbol{\alpha}_2(t) \end{pmatrix} \tag{14}$$

is a $2n_s x 2m_s$ time-dependent matrix

$$\overline{Z}_s = \begin{pmatrix} \overline{X}_s & \overline{\dot{X}}_s \end{pmatrix}^T \tag{15}$$

is the structural space vector. Equations for the space state structure can be summarized as

$$\overline{\dot{Z}}(t) = A(t)\overline{Z}(t) + G(t)\overline{W}_f(t) \tag{16}$$

where

$$\overline{Z}^T = \begin{pmatrix} \overline{X}_s, \overline{X}_f, \overline{\dot{X}}_S, \overline{\dot{X}}_f \end{pmatrix}^T \tag{17}$$

$$\mathbf{G}(t) = \begin{pmatrix} \mathbf{0} \\ \mathbf{G}_f \end{pmatrix} \tag{18}$$

is a new global $2m = 2\left(m_s + m_f\right)$ space state vector (structure plus filter) and

$$\frac{d}{dt}\begin{pmatrix} \overline{X}_s \\ \overline{X}_f \\ \dot{X}_S \\ \overline{\dot{X}}_f \end{pmatrix} = \begin{pmatrix} 0 & 0 & I & 0 \\ 0 & 0 & 0 & I \\ H_s^1 & G_s\alpha_1(t) & H_s^2 & G_s\alpha_2(t) \\ 0 & H_f^1(t) & 0 & H_f^1(t) \end{pmatrix} \begin{pmatrix} \overline{X}_s \\ \overline{X}_f \\ \dot{X}_S \\ \overline{\dot{X}}_f \end{pmatrix} + \begin{pmatrix} \overline{0} \\ \overline{0} \\ \overline{0} \\ G_f\overline{W}_f \end{pmatrix} \tag{19}$$

The structural matrix and response vectors are contingent on the design parameter vector $\bar{b}$, which encompasses elements such as structural stiffness, damping, masses, and various mechanical parameters like cross-sections, Young's modulus, and boundary conditions, among others. Filter parameters and input intensity are also included within this set of design parameters. Consequently, the system matrix and equations, comprising both the space state structure and filter equation, can be explicitly reconfigured as a function of this design parameter vector:

$$\mathbf{A}(b,\,t) = \begin{pmatrix} \mathbf{0} & \mathbf{0} & \mathbf{I} & \mathbf{0} \\ \mathbf{0} & \mathbf{0} & \mathbf{0} & \mathbf{I} \\ \mathbf{H}_s^1(\bar{b}) & \mathbf{G}_s\boldsymbol{\alpha}_1(\bar{b},\,t) & \mathbf{H}_s^2(\bar{b}) & \mathbf{G}_s\boldsymbol{\alpha}_2(\bar{b},\,t) \\ \mathbf{0} & \mathbf{H}_f^1(\bar{b},\,t) & \mathbf{0} & \mathbf{H}_f^1(\bar{b},\,t) \end{pmatrix} \tag{20}$$

$$\dot{\overline{Z}}(b,\,t) = A(\bar{b},\,t)\overline{Z}(\bar{b},\,t) + G(\bar{b},\,t)\overline{W}_f(t) \tag{21}$$

## 3. Space State Covariance Evaluation

In case of zero initial conditions, the solution of (space state structure + filter equation) has the following general expression:

$$\overline{Z}(t) = \int_0^t \boldsymbol{\Phi}(t,\,\tau)G\overline{W}_f(t)d\tau \tag{22}$$

where the matrix $\boldsymbol{\Phi}(t_1,\,t_2)$ (see for example [3]) is usually called *transition matrix*. The mean space state vector $\overline{\mu}_z(t)$ could be determined by the differential vectorial equation:

$$\dot{\overline{\mu}}_z\left(\bar{b},\,t\right) = A\overline{\mu}_z\left(\bar{b},\,t\right) + G\overline{\mu}_w(t) \tag{23}$$

The covariance matrix is as follows:

$$\mathbf{R}_{ZZ}(t_1,\,t_2) = \begin{pmatrix} \mathbf{R}_{XX}(t_1,\,t_2) & \mathbf{R}_{X\dot{X}}(t_1,\,t_2) \\ \mathbf{R}_{X\dot{X}}(t_1,\,t_2) & \mathbf{R}_{\dot{X}\dot{X}}(t_1,\,t_2) \end{pmatrix} \tag{24}$$

This second-order statistical moments matrix, due to its symmetry, is described by $(2m^2 + m)$ independent elements and can be evaluated by the well-known Lyapunov differential matrix equation

$$\dot{R}_{ZZ}(\bar{b},\,t) = A(\bar{b},\,t)R_{ZZ}(\bar{b},\,t) + R_{ZZ}(\bar{b},\,t)A(\bar{b},\,t)^T + B(t) \tag{25}$$

where

$$\mathbf{B}(t) = \left\langle \overline{Z}(t)\mathbf{G}^T\overline{W}(t)^T \right\rangle + \left\langle \mathbf{G}\overline{W}(t)\overline{Z}^T(t) \right\rangle = \mathbf{P}(t) + \mathbf{P}^T(t) \tag{26}$$

and where **P** can be written as

$$\mathbf{P}(t) = \int_0^t \boldsymbol{\Phi}(t-\tau)\mathbf{G}\left\langle \overline{W}(\tau)\overline{W}^T(t) \right\rangle \mathbf{G}^T d\tau = \int_0^t \boldsymbol{\Phi}(t-\tau)\mathbf{N}(t,\,\tau)d\tau \tag{27}$$

and

$$\mathbf{N}(t,\,\tau) = \mathbf{G}\mathbf{R}_{ww}(t,\,\tau)\mathbf{G}^T. \tag{28}$$

It must be noticed that Equation (25), which is valid for the non-stationary case, in a stationary environmental situation, is simplified in the following equation that contains no more time dependency:

$$A\left(\bar{b}\right)R_{ZZ}\left(\bar{b}\right) + R_{ZZ}\left(\bar{b}\right)A\left(\bar{b}\right)^T + B = 0 \tag{29}$$

A serious simplification takes place when the forcing vector is a white noise process as defined in (Rww stationary) or (Rww non-stationary). In these cases, the matrix $\mathbf{B}(t)$ is equal to

$$\mathbf{B}(t) = \left( \int_0^t \mathbf{\Phi}(t - \tau)\mathbf{G}\mathbf{R}_{ww}(t, \tau)\mathbf{G}^T d\tau \right) + \left( \int_0^t \mathbf{\Phi}(t - \tau)\mathbf{G}\mathbf{R}_{ww}(t, \tau)\mathbf{G}^T d\tau \right)^T \quad (30)$$

where both integrals above are equal due to the Dirac function properties. Finally, $\mathbf{B}$ can be written as follows:

$$\mathbf{B}(t) = 2\mathbf{G}\mathbf{R}_{ww}(t, t)\mathbf{G}^T = \begin{bmatrix} \mathbf{0}^{mxm} & \mathbf{0}^{mxm} \\ \mathbf{0}^{mxm} & \mathbf{L}(t) \end{bmatrix} \quad (31)$$

where the MXM submatrix is diagonal with the elements

$$[\mathbf{L}(t)]_{k,k} = \frac{2\pi S_{0_k}}{m_k^2} \varphi_k^2(t) \quad (32)$$

Meanwhile, for some applications, covariance information about structural acceleration is needed so that the matrix

$$\mathbf{R}_{\overline{\ddot{X}\ddot{X}}}(b, t) = \left\langle \overline{\ddot{X}\ddot{X}}^T \right\rangle \quad (33)$$

must be determined, and it is easily obtainable by the relation

$$R_{\overline{\ddot{X}\ddot{X}}}(\overline{b}, t) = D(b, t)R_{\overline{ZZ}}(b, t)D(b, t)^T \quad (34)$$

where

$$D(b, t) = \begin{bmatrix} \mathbf{H}_s^1(b) & \mathbf{G}_s\boldsymbol{\alpha}_1(b, t) & \mathbf{H}_s^2(b) & \mathbf{G}_s\boldsymbol{\alpha}_2(b, t) \end{bmatrix}. \quad (35)$$

## 4. Time Integration BF Procedure for R and $\mathbf{R}_{,\overline{b}}$

Even if many numerical standard codes exist for the stationary Lyapunov equation (the Lyapunov equation in stationary conditions is where A and B are the input matrices and X is the unknown one), there are limited examples for addressing the non-stationary Lyapunov equation, and so a simple numeric implicit integration method is proposed. In this context, a straightforward numerical implicit integration method is suggested. Specifically, the modified *Euler* method is used, in which the period is divided in $m$ equals time steps d in each sub-period $\Delta t$, and a linear variation in the time derivative covariance matrix $\dot{R}(t)$ is assumed. Under this assumption, we have the (*standard implicit Euler method*):

$$R^{(h+1)} = R^{(h)} + \frac{1}{2}\Delta t \left[ \dot{R}^{(h+1)} + \dot{R}^{(h)} \right] \quad (36)$$

where the symbol $a^{(h)}$ denotes the generic quantity $a$ evaluated at time $t = h\Delta t$. By using the matrix equations evaluated at times $t^{(h+1)}$ and $t^{(h)}$, we obtain the following $m$ algebraic matrix equations of the Lyapunov type

$$\left[ \left( \tfrac{1}{2}(\mathbf{I} - \Delta t\mathbf{A}) \right) \mathbf{R}^{(h+1)} + \mathbf{R}^{(h+1)} \left( \tfrac{1}{2}(\mathbf{I} - \Delta t\mathbf{A}) \right)^T \right] = $$
$$\left[ \left( \tfrac{1}{2}(\mathbf{I} + \Delta t\mathbf{A}) \right) \mathbf{R}^{(h)} + \mathbf{R}^{(h)} \left( \tfrac{1}{2}(\mathbf{I} + \Delta t\mathbf{A}) \right)^T \right] + \tfrac{\Delta t}{2} \left( \mathbf{B}^{(h)} + \mathbf{B}^{(h+1)} \right) \quad (37)$$

that are solved in sequence for each time value, starting from the initial time value and the initial covariance matrix value.

In this way, the $m$ unknown matrices ($h = 1, \ldots, m$) are determined. By assuming a constant or time variable (depending on the filter parameters variation), the matrices are

$$\mathbf{P}_B = \tfrac{1}{2}(\mathbf{I} - \Delta t\mathbf{A})$$

$$\mathbf{P}_F = \tfrac{1}{2}(\mathbf{I} + \Delta t\mathbf{A}) \quad (38)$$

and we discern that a more compact form of (Rnum1) is

$$\left[ \mathbf{P}_B \mathbf{R}^{(h+1)} + \mathbf{R}^{(h+1)} \mathbf{P}_B^T \right] = \left[ \mathbf{P}_F \mathbf{R}^{(h)} + \mathbf{R}^{(h)} \mathbf{P}_F^T \right] + \frac{\Delta t}{2} \left( \mathbf{B}^{(h)} + \mathbf{B}^{(h+1)} \right) \tag{39}$$

that could be solved at each step via a standard stationary Lyapunov equation solver, for example, the leap in the standard Matlab toolbox, in the form

$$\mathbf{P}_B \mathbf{R}^{(h+1)} + \mathbf{R}^{(h+1)} \mathbf{P}_B^T + \mathbf{C}^{(h+1)} = \mathbf{0} \tag{40}$$

where $\mathbf{C}^{(h+1)} = -\left( \left[ \mathbf{P}_F \mathbf{R}^{(h)} + \mathbf{R}^{(h)} \mathbf{P}_F^T \right] + \frac{\Delta t}{2} \left( \mathbf{B}^{(h)} + \mathbf{B}^{(h+1)} \right) \right)$.

Covariance could be integrated as proposed in the following integration scheme (Algorithm 1), where $n_t$ is the number of time integration steps, $T$ is the total analysis time, $n_b$ the number of design parameters, and, for the sake of simplicity in notation, $\mathbf{S}(h, j) = \mathbf{R}_{,b_j}((h-1)\Delta t)$:

---

**Algorithm 1:** Integration scheme

---

% $data\ input\left( \mathbf{M}, \mathbf{C}, \mathbf{K}, S_0, \omega_f, \xi_f, \phi(t) \right)$ %

$\mathbf{P}_B = \frac{1}{2}\left( \mathbf{I} - \Delta t \mathbf{A} \right)$

$\mathbf{P}_F = \frac{1}{2}\left( \mathbf{I} + \Delta t \mathbf{A} \right)$

$\mathbf{R}(1) = \mathbf{0};$      *(definition of initial condition for t=0 both on* $\mathbf{R}$ *and on* $\mathbf{R}_{,b}$ *)*

*for*   $j = 1$   *to*   $n_b$   % *initial condition definition*%

     $\mathbf{S}(j,1) = 0$ ;

*end;*

*for*   $i = 1$   *to* $(n_t - 1)$    % *beginning of time integration* %

       $\mathbf{N}(i) = \frac{\Delta t}{2}\left( \mathbf{B}(i+1) + \mathbf{B}(i) \right)$

       $\mathbf{C}(i) = \mathbf{P}_F \mathbf{R}(i) + \mathbf{R}(i)\mathbf{P}_F^T + \mathbf{N}(i)$

       $\mathbf{R}(i+1) = $ leap $(\mathbf{P}_B, -\mathbf{C}(i))$

       *standard solution for stationary Lyapunov equation*      $\mathbf{R}(i+1)\mathbf{P}_B + \mathbf{P}_B \mathbf{R}^T(i+1) - \mathbf{C}(i) = 0$.

       *for*   $j = 1$ *to* $n_b$    *cycle for each design vector element*

          $\mathbf{M}(i,j) = \frac{\Delta t}{2}\left[ \mathbf{A}_b(j)\left( \mathbf{R}(i) + \mathbf{R}(i+1) \right) + \left( \mathbf{R}(i) + \mathbf{R}(i+1) \right)\mathbf{A}_b^T(j) \right]$

          $\mathbf{D}(i,j) = \mathbf{P}_F \mathbf{S}(i,j) + \mathbf{S}(i,j)\mathbf{P}_F^T + \mathbf{M}(i,j)$

          $\mathbf{S}(i+1,j) = $ lyap $(\mathbf{P}_B, \mathbf{D}(i,j))$

       *end*     % *end of cycle on design vector elements* %

*end*    % *end of time integration* %

---

### 5. Numerical Example

The method proposed is used to analyze a multi-story building under earthquake forces. To maintain a balance between simplicity and generality, a shear-type plane frame structure is chosen as the model (as shown in Figure 1). This choice is reasonable because, in many buildings, the floor slabs possess very high in-plane stiffness, allowing them to be treated as rigid diaphragms. This simplification significantly enhances analysis efficiency without substantially compromising the accuracy of response assessment to ground forces.

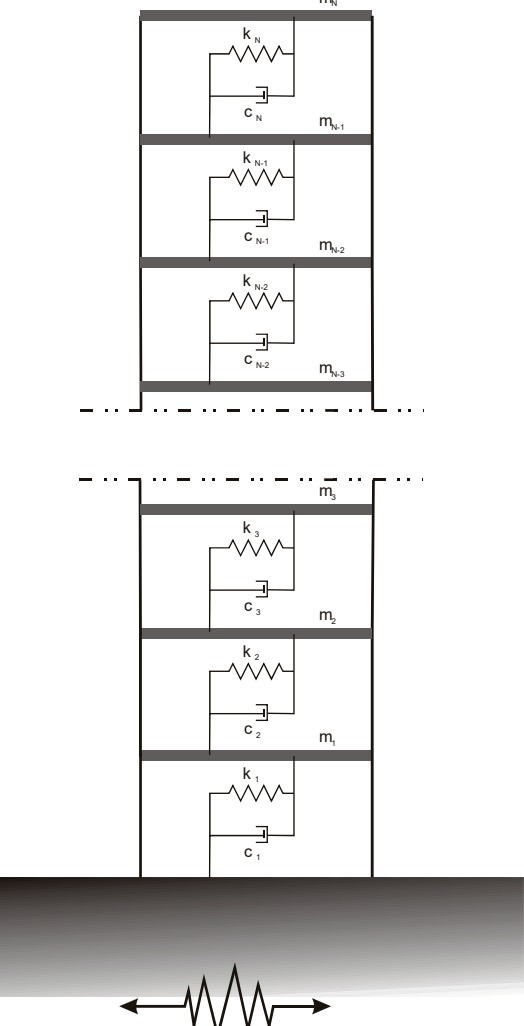

**Figure 1.** Mechanical scheme of analyzed plane frame shear type.

Moreover, to further enhance computational efficiency, the matrix condensation technique is employed. The primary assumption in modeling the building's mechanics is linearity, which remains valid considering the limitations on horizontal displacements necessary for operational service levels. This assumption holds when the maximum inter-story drift approaches or reaches the elastic limit of structural displacements, typical in full operational design demands.

For the sensitivity analysis, the design vector comprises the masses, stiffness, and damping of each floor. This comprehensive vector allows for the evaluation of how variations in these parameters impact the final structural reliability.

$$\bar{b} = (\bar{b}_1, \bar{b}_2, \bar{b}_3) = \left( \bar{m}^T, \bar{k}^T, \bar{c}^T \right)$$

where $\bar{b}_2 = \bar{k}^T = (k_1, k_2, k_3, \ldots, k_n)$.

### 5.1. Equations of Motion

To capture the essential seismic characteristics involving spectral and time modulation, a non-stationary modulated Kanai–Tajimi process is utilized to model stochastic ground motion. This process characterizes the base acceleration $\ddot{X}_g(t)$ acting at the structure's base as follows:

$$\begin{cases} \ddot{X}_g(t) = \ddot{X}_f(t) + \phi(t)w(t) \\ \ddot{X}_f(t) + 2\xi_g\omega_g\dot{X}_f(t) + \omega_g^2 X_f(t) = -\phi(t)w(t) \end{cases}$$

where $X_f(t)$ is the response of the Kanai–Tajimi filter, with a frequency $\omega_g$ and damping coefficient $\xi_g$, and $w(t)$ is the white noise, whose constant bilateral power spectral density (PDS) function is $S_0$. This last parameter is related to the peak ground acceleration (PGA) $\ddot{X}_g^{max}$ by means of relation [16]

$$S_0 = 0.2222 \frac{\xi_g \left(\ddot{X}_g^{max}\right)^2}{\pi\omega_g\left(1 + 4\xi_g^2\right)}$$

The non-stationary nature is introduced through the deterministic temporal modulation function $\phi(t)$, regulating the intensity variations while preserving the earthquake's frequency characteristics. In this scenario, the specific modulation functions proposed by Jennings [17] are adopted:

$$\varphi(t) = \begin{cases} \left(\dfrac{t}{t_1}\right)^2 & t < t_1 \\ 1 & t_1 \leq t \leq t_2 \\ e^{-\beta(t-t_2)} & t > t_2 \end{cases} \tag{41}$$

The motion equations for the complete structural system are then as follows:

$$\ddot{\overline{X}}(t) + M(\overline{b})^b C(\overline{b})\dot{\overline{X}}(t) + M(\overline{b})^b K(\overline{b})\overline{X}(t) = \overline{r}\ddot{X}_g(t)$$
$$\ddot{X}_f(t) + 2\xi_f\omega_f\dot{X}_f(t) + \omega_f^2 X_f(t) = -w(t)\phi(t)$$

where the drag vector $nx1$ is $\overline{r} = [1, 1, 1, \ldots, 1, 1]^T$, $\mathbf{M}(\overline{b})$, $\mathbf{C}(\overline{b})$, and $\mathbf{K}(\overline{b})$ are, respectively, mass, viscosity, and stiffness $n_x n$ principal structure matrices, whose general expression are reported in Appendix A with reference to the rigid floor assumption. The three vectors $\ddot{\overline{X}}(\overline{b}, t)$, $\dot{\overline{X}}(\overline{b}, t)$, and $\overline{X}(\overline{b}, t)$ are ground relative acceleration, velocity, and displacement $n_x 1$ vectors. In the case of the analyzed structure, the mass matrix is diagonal and the two viscous and stiffens matrices are tri-diagonal once. The mechanical filter parameters, the damping ratio and frequency, are $\xi_f$ and $\omega_f$, and the base excitation $\ddot{X}_g(t)$ is then equal to $\phi(t)w(t) + \ddot{X}_f(t)$.

Introducing the state vector $\overline{Y}(\overline{b}, t) = \left\{\overline{X}^T(\overline{b}, t), X_f(t)\right\}^T$, the motion equation system (motion Equation (1)) can be rewritten as

$$\ddot{\overline{Y}}(\overline{b}, t) = -H_1(\overline{b})\dot{\overline{Y}}(\overline{b}, t) - H_2(\overline{b})\overline{Y}(\overline{b}, t) - \overline{f}(t)$$

where the two matrices $\mathbf{H}_1(\overline{b})$ and $\mathbf{H}_2(\overline{b})$, with the vector $\overline{f}(t)$, are defined for this specific problem and shown in Appendix B.

The $N = n + 1$ degree of freedom 2nd-order differential system (second-order moment equals complete) can be replaced with a $2N$ DoF 1st-order differential equation in the space state as follows:

$$\dot{\overline{Z}}(\overline{b}, t) = A(\overline{b})\overline{Z}(\overline{b}, t) + \overline{F}(t)$$

where the space vector $2N$ is $\overline{Z}(\overline{b}, t) = \left\{ \overline{Y}(\overline{b}, t), \dot{\overline{Y}}(\overline{b}, t) \right\}$ and the *system matrix* (2N X. 2N)

$$\mathbf{A}(\overline{b}) = \begin{pmatrix} \mathbf{0}^{n+1} & \mathbf{I}^{n+1} \\ -\mathbf{H}_2(\overline{b}) & -\mathbf{H}_1(\overline{b}) \end{pmatrix} \text{ and the } 2N \text{ forcing vector } \overline{F}(t) = \left\{ \begin{matrix} \overline{0}^{n+1} \\ \overline{f}(t) \end{matrix} \right\}.$$

### 5.2. Reliability Evaluation

With reference to the proposed problem, it is required to evaluate the probability that each story drift $U_h$ of the floor exceeds the thresholds, at least once in a given earthquake duration. Then, for each hth level, this failure event is associated with the condition $|x_{h+1} - x_h| = |u_h| = \beta_h$. For each level $h$, the reliability vector element $r_h(\overline{b}, T)$ is defined as where, under the Poisson hypothesis for threshold crossing (that is an acceptable hypothesis for rare events as in [18]), we obtain

$$r_h(\overline{b}, \beta_h, T) =$$

$$\exp\left\{ -\frac{1}{\pi} \int_0^T \left( \frac{\sigma_{\dot{U}_h}(\overline{b}, \tau)}{\sigma_{U_h}(\overline{b}, \tau)} \sqrt{1 - \rho_{U_h \dot{U}_h}^2(\overline{b}, t)} \exp\left\{ -\frac{1}{2}\eta_h^2(\overline{b}, \beta_h, \tau) \right\} \chi\left[ d_{U_h}(\overline{b}, \beta_h, t) \right] \right) d\tau \right\}$$

and the final global structural reliability is then

$$r_{global}(\overline{b}, \overline{\beta}, t) = \prod_{h=1}^{n} r_h\left(\overline{b}, \beta_h, t\right)$$

Although exact analytical solutions for this are generally unavailable, it is known that the equation (approximate upper-bound global reliability) provides an approximate, upper-bound estimate, as stated above, and can be used for design and pre-design purposes from a practical viewpoint, as in this study. In order to evaluate the reliability vector (adopting the Poisson approach) related to the inter-floor relative displacement threshold crossing, one needs to introduce the inter-story drift vector with the associated covariance matrix $R_{Z_U Z_U}(t)$ (see Appendix C).

The reliability vector $\overline{r}_U$ previously defined can be evaluated as the collection of

$$r_{Uh}(T) = r_0 \, e^{-\int_0^T v_{Uh}^+(\tau)d\tau}$$

where $v_{Uh}^+(\eta_{V_h})$ is a function of $\sigma_{U_h}^2$, $\sigma_{\dot{U}_h}^2$, and $\rho_{U_h \dot{U}_h}$, accordingly, and are, respectively, the $h$ and the $n + h$ diagonal elements of $\mathbf{R}_{Z_U Z_U}(t)$ and $\eta_{U_h} = \frac{\beta_h}{\sigma_{U_h}}$, with $\beta_h$ being the hth barrier.

The equation represents the probability that the inter-story displacement will cross the maximum acceptable value $\beta_h$ during the time interval $[0, T]$. All the barriers can be collected in the barrier vector. Its elements are assumed constant and equal to 3.0 cm for each floor, that is, there is a lateral drift equal to 1.0% in the case of inter-story height of 3 m.

### 5.3. System Parameters

The chosen building configuration consists of three stories, each with a uniform mass equal to x $10^5$ (kg) for each level. Lateral stiffness to the first floor is present, and it is assumed that a linear reduction decreases its value to the at the top floor ($k_2 = 5.1 \cdot 10^7$(N/m) and $k_3 = 4.2 \cdot 10^7$(N/m)). Finally, damping is evaluated by setting $c_i = 2\sqrt{m_i k_i}$ ($c_i = [3.0 \ 2.8 \ 2.5] \ 10^5$(N s/m)).

The seismic characterization is based on a peak ground acceleration (PGA) of 0.45 (g), and four distinct total durations (10, 20, 30, and 40 s) are employed. Figure 2 displays the structural inter-story covariances, measured in terms of displacement (a) and velocity (b). It is noteworthy that for durations exceeding this, structural responses attain a stationary level. Moreover, for this particular structural configuration, the covariance response of the first level is greater than the other two, even though the third is approximately half, while the second is only slightly smaller.

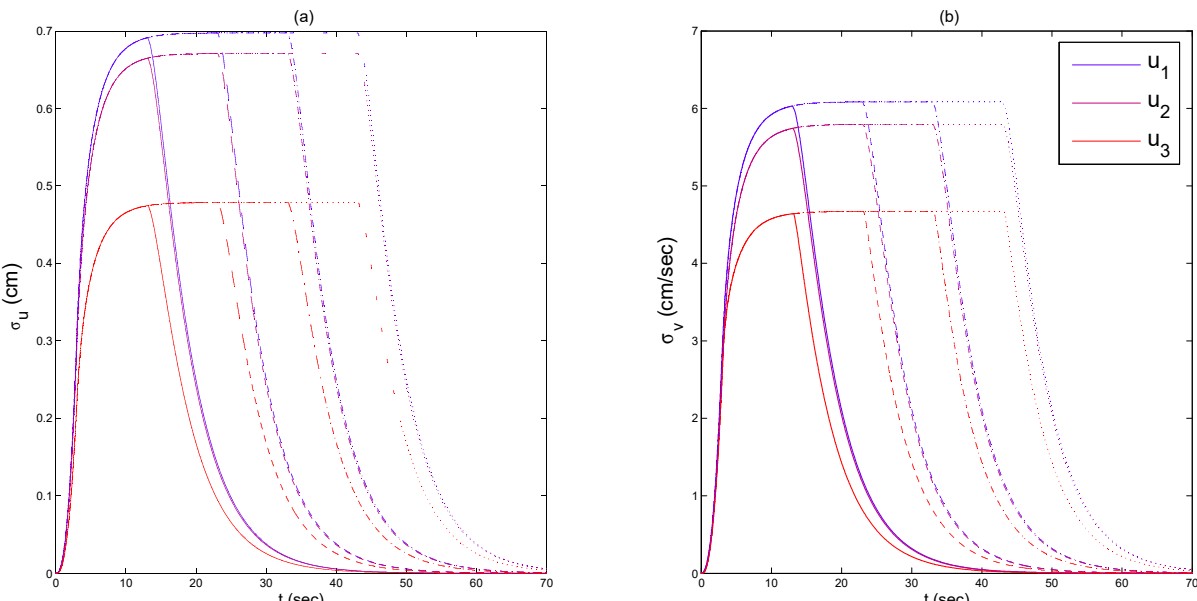

**Figure 2.** Inter-story displacements (**a**) and velocities (**b**) covariance response of a 3DoF system subject to modulated filtered white noise with different time durations. Continuous lines are for t_{d} = 10 (s), dashed lines are for t_{d} = 20 (s), dash–dot lines are for t_{d} = 30 (s), and finally, dotted lines are for t_{d} = 40 (s). Blue lines represent the inter-story drift response on the first floor, magenta lines on the second floor, and red lines refer to the third floor.

Figure 3 illustrates the structural safeties assessed at each lateral inter-story drift threshold for failure, along with the overall reliability calculated as an approximate upper-bound global reliability. It is important to observe that the probability of failure is highest for the first inter-story drift threshold, followed by a slightly lower probability for the second one, and finally, the third threshold has a considerably negligible probability of failure (i.e., $r_3 = 1$).

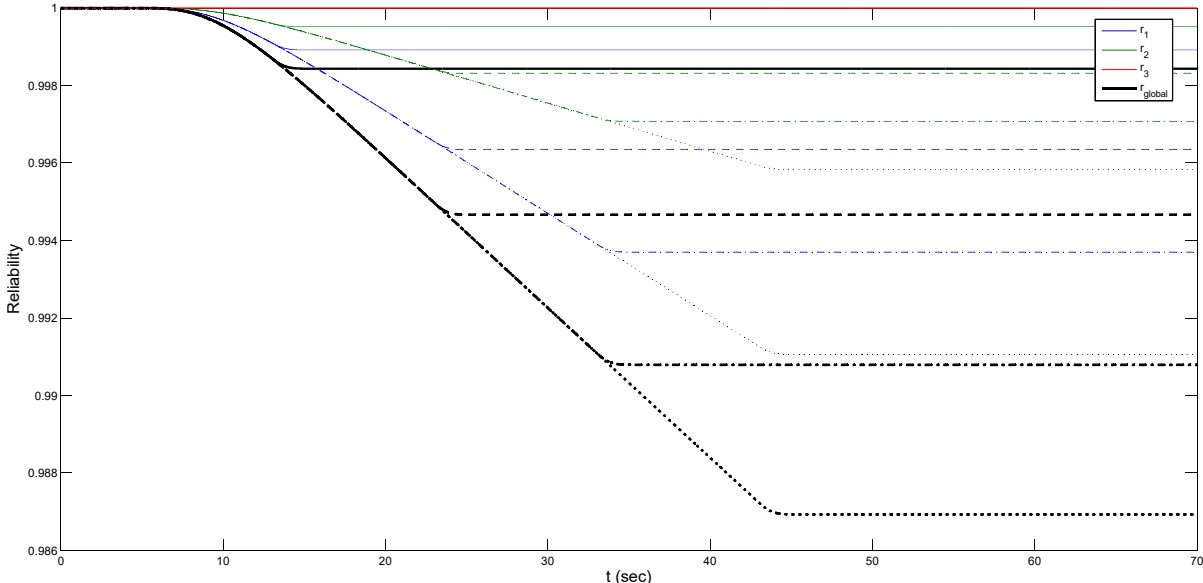

**Figure 3.** System reliability of a 3DoF system, evaluated as the probability of maximum inter-story drift exceeds a given threshold of 3 (cm). Results are obtained for different values of t_{d}: continuous lines are for t_{d} = 10 (s), dashed lines are for t_{d} = 20 (s), dash–dot lines are for t_{d} = 30 (s), and finally, dotted lines are for t_{d} = 40 (s). Blue lines are for first-level reliability, magenta lines are for second-level reliability, and red lines are for third-level reliability. Black slight lines are for global system reliability.

## 6. Conclusions

A numerical time integration algorithm is proposed to deal with non-stationary random vibration problems utilizing a covariance approach. The algorithm is developed to solve the differential matrix equations that govern the evolution of the stochastic response of structures subjected to random inputs. To create a versatile non-stationary approach applicable in various contexts, the structural response is assessed through a covariance approach. The reliability concerning first-crossing failure events is then derived based on the knowledge of the evolving covariance matrix in the space state. The algorithm is proposed for a generic Gaussian input of filtered non-stationary processes, representing diverse real-world physical loads. To solve this problem, the time integration algorithm involves differentiating the Lyapunov equation using an adapted Euler implicit scheme, which can be easily implemented using standard tools in various programming codes. Finally, the proposed algorithm is applied to analyze the dynamic responses of a multistory building, idealized as a shear frame structure.

**Author Contributions:** Conceptualization, R.C. and G.C.M.; Methodology, M.D., R.C. and G.C.M.; Validation, R.C. and G.C.M.; Formal analysis, M.D., R.C. and G.C.M.; Investigation, G.C.M.; Writing—original draft, M.D., R.C. and G.C.M.; Writing—review & editing, M.D., R.C., S.L.L., M.M. and G.C.M.; Visualization, L.S.; Supervision, G.C.M. All authors have read and agreed to the published version of the manuscript.

**Funding:** The research leading to these results has received funding from the European Research Council under the grant agreement ID: 101007595 of the project ADDOPTML, MSCA RISE 2020 Marie Skodowska Curie Research and Innovation Staff Exchange (RISE).

**Conflicts of Interest:** The authors declare no conflict of interest.

## Appendix A

Matrices, **C** and **K** are, for a shear-type frame, respectively diagonal and tri-diagonals:

$$\mathbf{M} = \begin{pmatrix} m_1 & 0 & 0 & & & \\ 0 & m_2 & 0 & & & \\ & 0 & m_3 & \ddots & & \\ & & \ddots & \ddots & & \\ & & & 0 & m_{n-1} & 0 \\ & & & & 0 & m_n \end{pmatrix} \tag{A1}$$

$$\mathbf{K} = \begin{pmatrix} k_1 + k_2 & -k_2 & 0 & & & \\ -k_2 & k_2 + k_3 & -k_3 & 0 & & \\ 0 & -k_3 & k_3 + k_4 & \ddots & & \\ & 0 & \ddots & \ddots & -k_{n-1} & 0 \\ & & & -k_{n-1} & k_{n-1} + k_n & -k_n \\ & & & 0 & -k_n & k_n \end{pmatrix} \tag{A2}$$

$$\mathbf{C} = \begin{pmatrix} c_1 + c_2 & -c_2 & 0 & & & \\ -c_2 & c_2 + c_3 & -c_3 & 0 & & \\ 0 & -c_3 & c_3 + c_4 & -c_4 & & \\ & 0 & \ddots & \ddots & \ddots & 0 \\ & & & -c_{n-1} & c_{n-1} + c_n & -c_n \\ & & & 0 & -c_n & c_n \end{pmatrix} \tag{A3}$$

**Appendix B**

Matrices $\mathbf{H}_1$ and $\mathbf{H}_2$ are

$$\mathbf{H}_1 = \begin{pmatrix} & & & & 2\xi_f\omega_f \\ & & & & 2\xi_f\omega_f \\ & \mathbf{M}^{-1}\mathbf{C} & & & \cdot \\ & & & & \cdot \\ & & & & 2\xi_f\omega_f \\ & & & & 2\xi_f\omega_f \\ 0 & 0 & 0 & \ldots\ldots & 0 & 0 & -2\xi_f\omega_f \end{pmatrix} \tag{A4}$$

$$\mathbf{H}_2 = \begin{pmatrix} & & & & \omega_f^2 \\ & & & & \omega_f^2 \\ & \mathbf{M}^{-1}\mathbf{K} & & & \cdot \\ & & & & \cdot \\ & & & & \omega_f^2 \\ & & & & \omega_f^2 \\ 0 & 0 & 0 & \ldots\ldots & 0 & 0 & -\omega_f^2 \end{pmatrix} \tag{A5}$$

where $\omega_f^2$ and $\xi_f$ are filter characteristics and the forcing vector is $\overline{f}(t) = [0, 0, 0, \ldots\ldots, 0, \varphi(t)w(t)]^T$.

**Appendix C**

The covariance matrix $R_{Z_U Z_U}\left(\overline{b}, t\right)$ is defined in the linear space of stochastic processes as a linear equation related to the inter-story drift $\overline{U}(\overline{b}, t)$ and displacement $\overline{X}(\overline{b}, t)$ vectors, holds $U(\overline{b}, t) = \mathbf{T}\overline{X}(\overline{b}, t)$ where the transform matrix $\mathbf{T}$ is a bi-diagonal one, and is independent of the design vector:

$$\mathbf{T} = \begin{pmatrix} 1 & 0 & 0 & 0 & & & \cdots & & 0 \\ -1 & 1 & 0 & 0 & & & & & \\ 0 & -1 & 1 & 0 & & & & & \vdots \\ & & -1 & 1 & \ddots & & & & \\ & & & & \ddots & 1 & 0 & 0 & \\ \vdots & & & & 0 & -1 & 1 & 0 & 0 \\ & & & & & 0 & -1 & 1 & 0 \\ 0 & & & & \cdots & & & 0 & 1 \end{pmatrix} \tag{A6}$$

The covariance matrix $\mathbf{R}_{Z_U Z_U}\left(\overline{b}, t\right)$ is then related to $\mathbf{R}_{ZZ}\left(\overline{b}, t\right)$ through the following connection:

$$\widehat{T} = \begin{pmatrix} T & 0 \\ 0 & T \end{pmatrix} \tag{A7}$$

$$\overline{Z}_V(\overline{b}) = \widehat{T}\overline{Z}(\overline{b}) \tag{A8}$$

$$R_{Z_U Z_U}(\overline{b}, t) = \widehat{T}R_{ZZ}(\overline{b}, t)\widehat{T}^T \tag{A9}$$

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
