# Peer review of "Numerical Covariance Evaluation for Linear Structures Subject to Non-Stationary Random Inputs"

_computation, doi:10.3390/computation12030050_

Round 1

Reviewer 1 Report

Comments and Suggestions for Authors

This article proposed a numerical time integration algorithm for nonstationary random vibration analysis. The effectiveness and accuracy of the proposed method are validated by a numerical example. However, the paper still has the following shortages:

1.     This paper does not explicitly state whether the proposed method can only deal with white noise external excitation, please give a precise explanation in the problem statement part.

2.     The integration scheme shown after line 254 is hard to read, and there seems to be a for loop without an end. Please modify this this pseudocode part to improve its readability.

3.     Please explain the reasonability of obtaining the floor reliability rh (b, beta, T) based on the Poisson hypothesis in section 5.2.

4.     In section 5.2, the global structural reliability is commutated as the product of the reliability of all floors. This calculation method ignores the correlation between the failure events of each floor.

5.     Several relative research works are suggested: https://doi.org/10.1016/j.apm.2023.05.022; https://doi.org/10.1016/j.strusafe.2022.102313; https://doi.org/10.1016/j.ymssp.2023.110714.

6.     There are still some grammatical mistakes and spelling errors in the paper that need to be modified. For example, one comma on page 8, line 254, is redundant.

    Above all, although the investigation has advantages in practical implementation, some details should be considered carefully. The language of the paper should be modified carefully. I suggest the paper can be published after major revision.

Author Response

Dear authors, 

In the attachement, you can find the word document with the point-by-point response to the reviewer.

Reviewer 2 Report

Comments and Suggestions for Authors

This study is worth considering in the mathematical solving process of the response covariance in the time domain of a linear structural system under nonstationary stochastic loads, but several issues related to the mechanical field should be addressed before revisiting it.

[1] Issues with Inverse Matrices: Both matrices, H_1(b) and H_2(b) in Appendix B, consist of inverse matrices. The proposed mass matrices were simple diagonal matrices, simplifying the calculation process. However, this form may not effectively represent more complex systems. Linear mechanical systems can efficiently handle modal formulations using the orthogonal properties of eigenvectors.

[2] The highlighted point of the study focused on the numerical calculation process of the simplified theoretical model, but this was not well explained in the example chapter. The simulation results (Figure 3 and 4) do not reveal the claimed issues.

[3] The time-domain response signal for the simplified linear mechanical system has been well calculated in previous studies. It may be more efficient to solve it in the frequency domain, even with non-stationary input data. Therefore, the current literature review should be revised to account for relevant previous studies.

[4] Are the simulation results correct? They should be verified with other independent simulation results, such as those obtained from commercial software.

My opinion on the current version is negative.

Comments on the Quality of English Language

Extensive editing of English language required. 

Author Response

Dear editors and reviewers,

in the attachment, you can find the point-by-point response to the reviewer. 

Reviewer 3 Report

Comments and Suggestions for Authors

In this paper, to achieve a versatile non-stationary approach applicable across different contexts, the time integration algorithm involves differentiating the Lyapunov equation using an adapted Euler implicit scheme, which can be easily implemented using standard tools in various programming codes. The proposed algorithm is applied to analyze the dynamic responses of a multistory building, idealized as a shear frame structure. I feel that this work is interesting and well-performed. A numerical time integration algorithm is proposed to deal with nonstationary random vibration problems utilizing a covariance approach. The excellent performance and promising application of the accelerometer are satisfactory for publishing on computation. however, some errors need to be modified.

1. All formulas in the text need to be labeled

2, Figure 2, from "1" to "N", the transition process in the middle should be an ellipsis.

3. What do u1, u2 and u3 represent in Figure 3? It needs to be clearly labeled. The abscissa of the curve goes to 0 after 40 secs. Why does this happen at 40 secs? Perform a detailed analysis.

4, the author should give a detailed explanation of Figure 4 instead of a simple description.

5. English needs to be further optimized.

Comments on the Quality of English Language

Minor editing of English language required.

Author Response

(The authors gave the same response as above.)

Round 2

Reviewer 1 Report

Comments and Suggestions for Authors

Accept.

Reviewer 2 Report

Comments and Suggestions for Authors

The revised version is worthy to be considered in this journal. 

Comments on the Quality of English Language

Minor editing of English language required.